

# Practical guidance for firefighter applicants preparing for cardiorespiratory fitness testing: a secondary analysis of self-reported physical activity levels

Sylvie Fortier, Liam P. Kelly and Fabien A. Basset

School of Human Kinetics and Recreation, Memorial University of Newfoundland, St. John's, Newfoundland, Canada

## ABSTRACT

Adequate cardiorespiratory fitness is critical for firefighters since an insufficient level of fitness threatens the integrity of their operations and could be dangerous for their lives. In fact, the leading cause of mortality for on-duty firefighters is not injury but sudden cardiac death. Therefore, to mitigate these risks, potential firefighter recruits are often required to perform a graded exercise test to determine their cardiorespiratory fitness as part of the recruitment process. However, there are currently limited data available to prospective firefighters on the amounts and types of exercises needed to be successful in the graded exercise test, commonly known as a $\dot{V}O_{2max}$ test. Physiological parameters for the current secondary analysis were collected on firefighter applicants who performed the graded exercise test where 72% were successful and 28% were unsuccessful to meet the minimum standard set at 42.5 ml kg$^{-1}$ min$^{-1}$. Prior to their test, applicants were asked to describe their exercise training routine by indicating the number of minutes per week spent exercising. Activities were then divided into one of two categories: endurance exercise or strength and power exercise training. The total exercise training describes the sum of all activities performed each week. The sum of endurance exercise activities and the sum of strength and power exercise activities were compared between the successful and the unsuccessful groups and results showed that successful applicants had a higher training volume and performed more endurance exercise training as compared to unsuccessful applicants. Therefore, practical recommendations related to exercise training regime are presented for firefighter applicants to embrace as guidance to prepare for their graded exercise test as part of their recruitment process.

Corresponding author
Fabien A. Basset, fbasset@mun.ca

## INTRODUCTION

Maintaining cardiorespiratory fitness is essential for emergency responders as this factor might seriously impact on the integrity of their operations. Firefighters represent a unique occupational group in that regard. The physiological burden of wearing bulky protective clothing, breathing through a self-contained breathing apparatus, and performing

physically demanding work at the incident scene requires stamina and strength. Firefighter work environments also contain chemicals such as carbon dioxide at acutely toxic levels that alters cardiac rhythm, probably as a result of the induced hypoxia (*Kristensen, 1989*). It is, therefore, not surprising that the leading cause of mortality for on-duty firefighters is not injury but sudden cardiac death, accounting for approximately 44% of USA career firefighters on-duty fatalities (*Donovan et al., 2009*; *Fahy, 2005*; *Smith, Barr & Kales, 2013*). While the exact mechanism for this remains unclear, low cardiorespiratory fitness has been associated with increased risk of cardiovascular disease-related events (*Donovan et al., 2009*).

The physical nature and cardiovascular strain imposed during firefighting has been well documented and have led to many recommendations for minimum requirements of cardiorespiratory fitness (*Barnard & Duncan, 1975*; *Gledhill & Jamnik, 1992*; *Lemon & Hermiston, 1977*). While no universal 'fitness-for-duty' minimum standard exists, several studies have recorded oxygen uptake ($\dot{V}O_2$) during simulated firefighting tasks and reported values ranging from 23.0 to 43.8 ml kg$^{-1}$ min$^{-1}$. Accordingly, recommendations for maximal oxygen uptake ($\dot{V}O_{2max}$) between 39.6 and 48.5 ml kg$^{-1}$ min$^{-1}$ are available in the literature (*Gledhill & Jamnik, 1992*; *Lemon & Hermiston, 1977*; *Sothmann et al., 1991*; *Williams-Bell et al., 2010*). In Canada, Gledhill & Jamnik (*Gledhill & Jamnik, 1992*) conducted a task analysis to detail the specific tasks conducted by incumbent firefighters. The aim was to evaluate the physical abilities related to these tasks, determine physically demanding ones, and determine the physiological requirements to complete these tasks. Based on this analysis, the most demanding tasks required a mean $\dot{V}O_2$ of 41.5 ml kg$^{-1}$ min$^{-1}$ which was, on average, equivalent to 85% of the participants $\dot{V}O_{2max}$. Accordingly, values between 42.5 and 45.0 ml kg$^{-1}$ min$^{-1}$ have been widely accepted across Canada where Newfoundland and Labrador adopted the value of 42.5 ml kg$^{-1}$ min$^{-1}$ as their minimum standard for firefighter applicants, which was used as the cutoff value in the current secondary analysis. The recruitment process to become a firefighter is very competitive in Canada and it consists of multiple phases including, but not limited to, a graded exercise test (GXT) to assess aerobic capacity and an occupational-specific physical test where muscular strength and power play a major role to achieve a positive outcome. Failure to succeed in any of these tests will result in the application being rescinded. Although achieving a $\dot{V}O_2$ value of 42.5 ml kg$^{-1}$ min$^{-1}$ should not be problematic for young and active Caucasian males (*Kaminsky, Arena & Myers, 2015*), it remains that many individuals need to prepare for the physically demanding tests as part of the recruitment process. The preparation mainly involves training the cardiorespiratory system and gaining muscular strength. The aim of the current project was therefore to determine the potential relationship between a positive cardiorespiratory outcome on the GXT and the type of exercise training regimen performed by the applicants. It further intends to provide practical guidance to future firefighter applicants preparing for their GXT.

## MATERIALS & METHODS

The current study employed a secondary data analysis of self-reported physical activity levels to investigate differences in exercise training routine (type and amount) between

firefighter applicants who successfully achieved the minimum recruitment criteria for the GXT performed on a treadmill and those who were unsuccessful. The original data were collected from male firefighter applicants living in Newfoundland and Labrador who performed a GXT as part of their recruitment process. All applicants were requested to include in their application package a document certifying that they had recently (*i.e.,* <3 months ago) met the minimum cardiorespiratory fitness standard through a recognized evaluation site. Thus, the dataset used for this secondary analysis includes 121 applicants who chose to perform their GXT at the exercise physiology clinic located on Memorial University of Newfoundland campus. Prior to their testing appointment, applicants completed a Physical Activity Readiness Questionnaire (PAR-Q) adapted from the Canadian Society for Exercise Physiology. Their self-reported physical activity and exercise habits were included in the dataset. All applicants consented to have their data used for research purposes and the regional Health Research Ethics Board approved the study protocol (HIC-10.201).

## Self-reported physical activity and exercise habits

The PAR-Q was used as a screening tool to reveal any contraindications to administering a maximal graded exercise test (see Addendum 1). The questionnaire included, inter alia, questions related to cardiovascular, respiratory, and metabolic problems. The extended version also included questions about current physical activity participation and exercise habits where a list of 34 common sport and recreational activities were provided. Applicants were asked (i) to identify if they were physically active in the past year, (ii) to identify which activities they were currently participating in on a regular basis, and (iii) to indicate the number of minutes per week (min/week) spent performing each activity. Applicants also had the opportunity to indicate if they were participating in any activity other than the ones listed. Information collected from the PAR-Q was reviewed upon arrival at the evaluation site to ensure that there were no contraindications to maximal exercise testing.

## Graded exercise test

All applicants performed a GXT to determine their maximal $\dot{V}O_2$ value. Prior to their appointment, applicants were instructed (i) to refrain from strenuous exercise at least 36 h prior to the test, (ii) to avoid eating and consuming alcohol at least four hours prior to the test, and (iii) to avoid caffeine, drugs, and smoking within the two hours period preceding the test. The GXT was performed on a motor driven treadmill and the Léger-Boucher protocol was used which involves a gradual increase in speed over time until exhaustion (*Léger & Boucher, 1980*). The GXT consisted of three phases: the warm-up phase, the exercise phase, and the cool-down phase. Prior to beginning the warm-up phase, the applicants spent two minutes on the treadmill breathing normally to collect physiological values in the standing position. The warm-up phase was five minutes in length and the treadmill was set to a walking speed of 4.5 km/h and 1% grade. The grade remained consistent at 1% throughout the test. Once the exercise phase had started, the speed of the treadmill increased to 7 km/h for the first two minutes and then increased by a rate of 1 km/h every two minutes thereafter until exhaustion. Verbal encouragement was

given to the applicants to exercise until they reached volitional exhaustion. The cool-down phase included walking at a speed of 4.5 km/h for a minimum of five minutes. During the cool-down phase, the exercise physiologist decided whether or not a verification phase was required and was safe to administer. A verification phase is normally used as a criterion to ensure that participants reached their 'true' maximum oxygen uptake during the incremental exercise phase (*Rossiter, Kowalchuk & Whipp, 2006*). However, since the purpose of the GXT was to provide applicants with a certificate stating that they reached the minimum standard for firefighters, the verification phase was not implemented to applicants who clearly surpassed a $\dot{V}O_2$ value of 42.5 ml kg$^{-1}$ min$^{-1}$ during the exercise phase. For applicants who performed the verification phase, the speed of the treadmill was increased to 105% of their maximal aerobic speed reached during the preceding exercise phase. The speed and the 1% grade remained the same throughout the entire verification period, which lasted until volitional exhaustion once again. For those who completed the verification phase, the highest value reached during any phase throughout the GXT was retained as their maximal $\dot{V}O_2$ value.

## Physiological measurements

Exercise metabolic rates were recorded with an indirect calorimetry system (Oxycon Pro; Jaeger, Hochberg, Germany) and heart rate (HR) was recorded using a heart rate monitor (Polar Electro Canada, Lachine, Qc, Canada). Oxygen uptake ($\dot{V}O_2$), carbon dioxide ($\dot{V}CO_2$), and breathing frequency (Bf) were recorded breath-by-breath *via* an automated open-circuit gas analysis system with $O_2$ fuel cell and $CO_2$ infared cell in connection a turbine flowmeter. Respiratory exchange ratio (RER) and minute ventilation ($\dot{V}_E$) were calculated as the quotient of $\dot{V}CO_2$ on $\dot{V}O_2$ and as the product of breathing frequency by tidal volume, respectively. Prior to testing, volume and gas analyzers were calibrated with a 3.0 L calibration syringe and certified $O_2$ and $CO_2$ calibration gases at 16% and 4%, respectfully. In addition, propane gas calibration was performed to ensure accuracy and linearity of the indirect calorimetry system according to *Ismail et al. (2019)* methodology. The data were online digitized from an A/D card to a computer for live monitoring of the physiological parameters.

## Data handling

Raw data from the GXT were smoothed over a 30-second moving average to determine applicants' maximal physiological values (*Robergs & Burnett, 2003*). The data from the section pertaining to physical activity and exercise habits were used to calculate total weekly activities performed by each applicant. There were five observations greater than 1020 min/week which was considered excessive for this population. Since the PAR-Q was filled out in February, all activities that could only be performed in the summer in Newfoundland and Labrador were removed on the basis that applicants did not answer the questionnaire correctly, that is, to only include activities currently performed on a regular basis. In addition, although physical activity plays an important role in the prevention and treatment of multiple chronic diseases, we were interested in the amount of exercise applicants performed in preparation for the GXT. Physical activity and exercise are often

used interchangeably, but these terms are not synonymous. Physical activity is defined as any bodily movement produced by the contraction of skeletal muscles that results in a substantial increase in caloric requirements over resting energy expenditure. Exercise is a type of physical activity consisting of planned, structured, and repetitive bodily movements organized in such a way as to improve and/or maintain one or more components of physical fitness (*Liguori, 2021*). Since the parameter of interest with firefighter applicants is exercise performed with the main objective to improve or maintain physical fitness, activities considered as physical activity (*e.g.*, walking and dancing) were also removed. The reader is invited to look at Addendum 2 for more details about data handling.

Next, all exercise activities were divided into one of two categories: endurance exercise or strength and power exercise. Also called aerobic exercise, the term endurance exercise training generally refers to training the aerobic system and includes activities that increase your breathing and HR. From the PAR-Q, 31 activities were considered endurance-based (*e.g.,* running, biking, hockey, etc.) whereas three were strength and power activities (*i.e.,* resistance training, downhill skiing, and martial arts). Strength and power exercises are commonly called resistance training, weight training, or power training, and improve muscular strength. For ease of readability, strength and power exercises will be referred to strength exercise for the remaining of the manuscript. The sum of all endurance exercises and all strength exercises in min/week were calculated for each applicant. The total exercise training was the addition of self-reported endurance exercises and self-reported strength exercises which describes the total training time or weekly training volume.

Finally, as an additional analysis, the original dataset was divided into three fitness zones based on maximal $\dot{V}O_2$ values obtained during the GXT. Maximal values on a GXT can vary from one day to the next. This variability may be partitioned into technological error and biological variation. Technological errors include the variable errors of the instruments, uncontrolled environment factors, reading errors, and unidentified errors. Yet, attempting to single out one or two major factors contributing to the biological variation in energy metabolism during maximal exercise is difficult (*Katch, Sady & Freedson, 1982*). The total variability is $\pm 5.6\%$ with biological variation accounting for $\geq 90\%$ of this variability, while technological error only accounts for $\leq 10\%$ (*Katch, Sady & Freedson, 1982*). In practical terms, it means that an applicant could have a higher or lower maximal $\dot{V}O_2$ value by 5.6% on any given day. Applicants should therefore aim for a minimum $\dot{V}O_2$ value $\geq 45$ ml $kg^{-1}$ $min^{-1}$ (*i.e.,* 42.5 + 5.6%) to be confident that they would succeed in the GXT even if they are tested on their least biologically optimal day. Correspondingly, the three fitness zones were divided according to the following $\dot{V}O_2$ values: $\leq 40$ (*i.e.,* 42.5–5.6%), 40 to 45, and $\geq 45$ (*i.e.,* 42.5 + 5.6%) ml $kg^{-1}$ $min^{-1}$. The middle fitness zone (40 to 45) is, in essence, a gambling zone because applicants could pass or fail the GXT depending on any combination of factors involved in the total variability on testing day.

## Statistical analysis

Normality was inspected by examining the histograms and conducting the Shapiro–Wilk test which showed that most parameters followed a normal distribution except for age and body mass. However, since the central limit theorem states that the distribution of sample

means approximates a normal distribution as the sample size gets larger than 30, normality was assumed for all parameters used for calculating the $t$ statistic.

The complete dataset was divided into two groups, successful group (SG) and unsuccessful group (UG), to assess if there was a statistical difference between the two groups. Two-sample Welch's $t$-test was conducted on all parameters using Python 3 (Python Software Foundation, Wilmington, DE, USA) to test the equality of two means from independent populations. Subsequently, a one-way Welch ANOVA was performed on parameters of interest to assess the difference between the mean for each of the three fitness zones and a pairwise Games-Howell post hoc test was used to compare all possible combinations of fitness zone differences.

Unstandardized as well as standardized effect size ($d$ statistic) are reported along with their corresponding confidence interval (CI) when applicable. Unstandardized effect size ($ES_{M1-M2}$) was calculated by subtracting the mean of UG from the mean of SG. Their CI were constructed using the standard error, $t$-score, and degree of freedom ($df$) for unequal variances as suggested by *Kline (2013)*. The standardized $d$ statistic effect size was calculated using the $t$ statistic (*Grissom & Kim, 2012*; *Kline, 2013*) and their CI were constructed using the standard error, $t$-score, and $df$ from the $d$ statistic (*Kline, 2013*). At last, conventional values proposed by *Cohen (1988)* were used as benchmarks for what are considered to be 'small', 'medium', and 'large' effects along with their potential effect (*Cohen, 1992*). For more details about statistical analysis, see Addendum 3.

## RESULTS

Table 1 tabulates the mean ($\pm$ SD) for age, height, body mass, and body mass index (BMI) for all applicants were as followed: (i) 26.3 ($\pm$ 5.8) years where the youngest applicant was 18 years old and the older was 44 years old, (ii) height of 178.6 ($\pm$ 6.2) cm where the shortest applicant was 160.0 cm and the tallest was 194.6 cm; (iii) body mass of 84.9 ($\pm$ 10.6) kg where the lightest applicant was 66.5 kg and the heaviest was 122.2 kg; and (iv) BMI of 26.6 ($\pm$ 2.7) ranging from 20.3 to 33.6.

When comparing the two groups (SG *vs.* UG), mean ($\pm$ SD), outcomes from the $t$ statistic, and the unstandardized effect size ($ES_{M1-M2}$) are presented in Table 1. Statistical significance was set at the 0.05 level of confidence. Out of 121 applicants, 87 (72%) were successful and 34 (28%) were unsuccessful in the GXT. Figure 1 displays the $d$ statistic effect size with the corresponding CI in relation to the effect size classes and potential effects. Figure 2 illustrates the difference in medians for body mass and HR recovery for each fitness zone mentioned earlier whereas Fig. 3 shows the median for endurance exercise and total exercise training time performed weekly by applicants for each of the three fitness zones. The one-way ANOVA test indicated that at least one fitness zone was statistically different and the post hoc test revealed that the red zone ($\leq 40$ ml kg$^{-1}$ min$^{-1}$) was statistically lower than the green zone ($\geq 45$ ml kg$^{-1}$ min$^{-1}$) for endurance exercise in minutes ($M_{red} = 109$, $M_{green} = 235$, $t$ (70.4) $= -4.5$, $p = .001$) and also for total exercise in minutes ($M_{red} = 268$, $M_{green} = 442$, $t$ (44.7) $= -3.5$, $p = .003$).

Fortier et al. (2022), *PeerJ*, DOI 10.7717/peerj.13832

**Table 1  Anthropometrics and physical characteristics of participants.**

| Parameters | N | Mean (all) (SD) | Mean SG (SD) | Mean UG (SD) | ES$_{M1-M2}$[95% CI] | Diff (%) | *t* statistic | *df* | *p* value |
|---|---|---|---|---|---|---|---|---|---|
| Age (years) | 121 | 26.3 (5.8) | 25.5 (4.9) | 28.4 (7.3) | −2.9 [−5.7, −0.1] | −10 | −2.091 | 44.7 | .042 |
| Height (cm) | 121 | 178.6 (6.2) | 178.4 (5.5) | 179.2 (7.7) | −0.8 [−3.8, 2.2] | 0 | −0.548 | 46.3 | .587 |
| Body mass (kg) | 121 | 84.9 (10.6) | 82.7 (8.3) | 90.4 (13.5) | −7.6 [−12.7, −2.5] | −8 | −3.022 | 43.0 | .004 |
| Maximal HR (bpm) | 121 | 195 (9) | 195 (9) | 194 (10) | 1 [−3, 5] | 0 | 0.363 | 56.4 | .718 |
| HR recovery (bpm) | 121 | 40 (10) | 42 (10) | 37 (10) | 5 [1, 9] | 14 | 2.554 | 61.7 | .013 |
| Maximal $\dot{V}O_2$ (ml kg$^{-1}$ min$^{-1}$) | 121 | 46.5 (6.4) | 49.6 (4.5) | 38.7 (2.7) | 10.9 [9.5, 12.2] | 28 | 15.896 | 97.6 | <.001 |
| Abs$\dot{V}O_2$ (ml min$^{-1}$) | 121 | 3928 (588) | 4099 (547) | 3491 (448) | 608 [413, 803] | 17 | 6.214 | 72.4 | <.001 |
| $\dot{V}CO_2$ (ml min$^{-1}$) | 119 | 3802 (571) | 3971 (529) | 3362 (425) | 609 [420, 797] | 18 | 6.438 | 71.1 | <.001 |
| $\dot{V}_E$ (L min$^{-1}$) | 119 | 130 (20) | 135 (18) | 116 (18) | 19 [11, 26] | 16 | 5.049 | 57.0 | <.001 |
| Bf (breath min$^{-1}$) | 119 | 50 (8) | 51 (8) | 48 (8) | 3 [0, 7] | 7 | 2.044 | 55.0 | .046 |
| RER | 119 | 1.0 (0) | 1.0 (0) | 1.0 (0) | 0 [0, 0] | 0 | 0.372 | 46.2 | .711 |
| EE (min/week) | 120 | 195 (154) | 216 (162) | 141 (117) | 75 [21, 128] | 53 | 2.779 | 82.7 | .007 |
| SE (min/week) | 119 | 189 (158) | 202 (159) | 156 (148) | 46 [−16, 108] | 30 | 1.481 | 64.5 | .143 |
| TE (min/week) | 118 | 382 (239) | 417 (240) | 298 (213) | 119 [28, 210] | 40 | 2.612 | 68.0 | .011 |

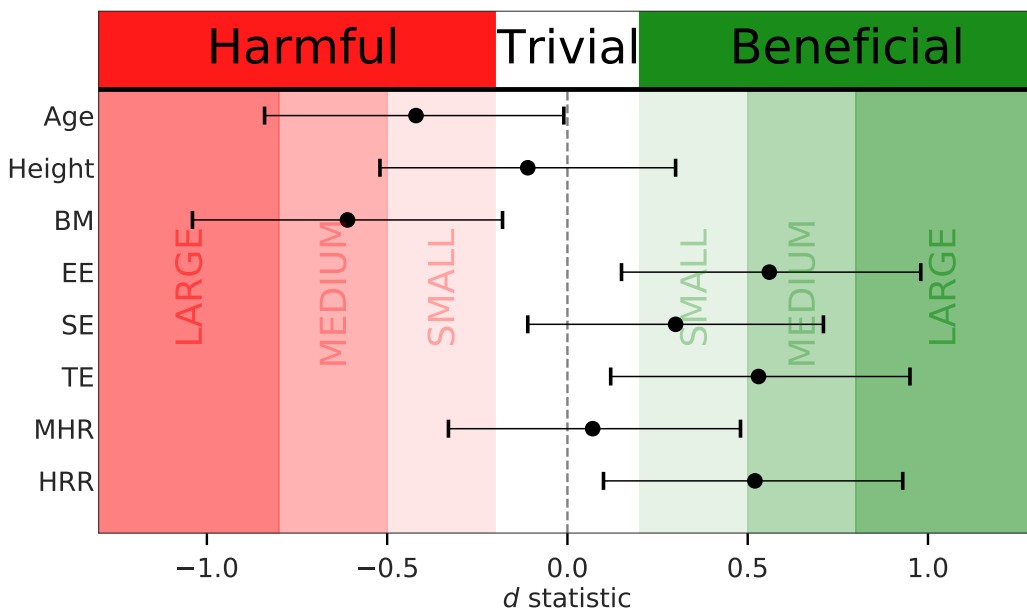

**Figure 1** Effect size and confidence intervals for *d* statistic in relation to effect size classes and potential effects. Adapted from *Cohen (1988)* and *Batterham & Hopkins (2006)*.

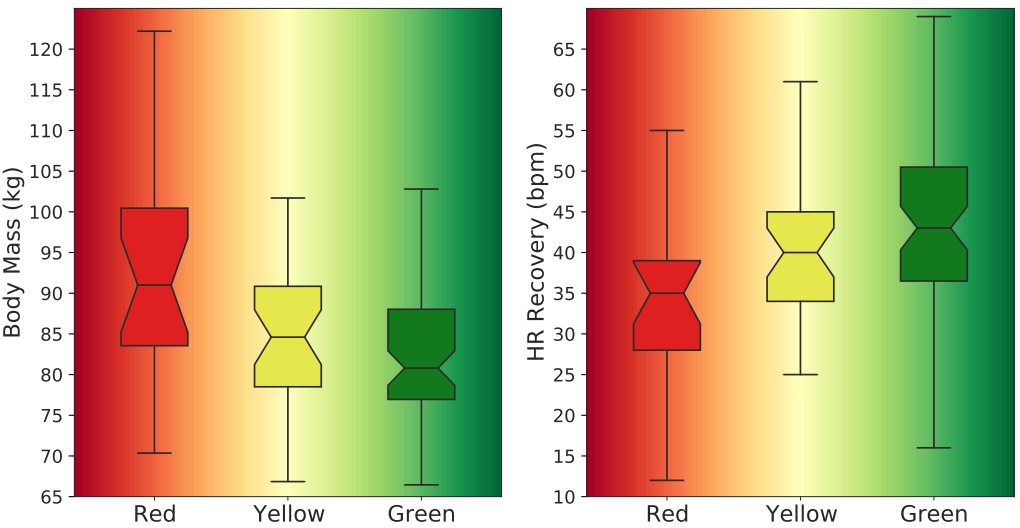

**Figure 2** Difference in medians for body mass and heart rate recovery for each fitness zone ($n_{red} = 21$, $n_{yellow} = 33$, $n_{green} = 67$). The green zone includes applicants with maximal $\dot{V}O_2$ values $\geq 45$ ml kg$^{-1}$ min$^{-1}$, the yellow zone includes applicants with maximal $\dot{V}O_2$ values between 40–45 ml kg$^{-1}$ min$^{-1}$, and the red zone includes applicants with maximal $\dot{V}O_2$ values $\leq 40$ ml kg$^{-1}$ min$^{-1}$.

## DISCUSSION

The project was undertaken to determine whether the type of exercise training regimen impacts the GXT score in individuals who prepare for the physically demanding tests as

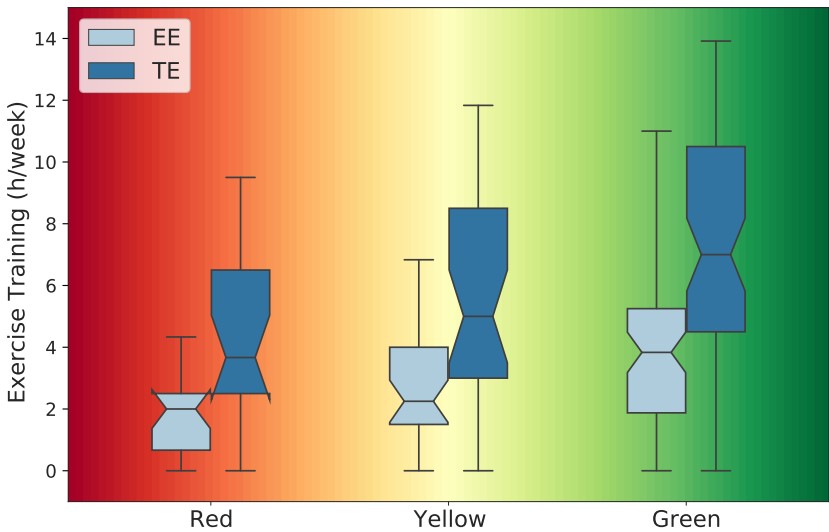

**Figure 3** **Difference in medians for endurance exercise (EE) and total exercise (TE) training time in hours per week for each fitness zone ($n_{red}$ = 21, $n_{yellow}$ = 33, $n_{green}$ = 67).** The green zone includes applicants with maximal $\dot{V}O_2$ values $\geq$ 45 ml kg$^{-1}$ min$^{-1}$, the yellow zone includes applicants with maximal $\dot{V}O_2$ values between 40–45 ml kg$^{-1}$ min$^{-1}$, and the red zone includes applicants with maximal $\dot{V}O_2$ values $\leq$ 40 ml kg$^{-1}$ min$^{-1}$.

part of the firefighter recruitment process. Our results confirm the relationship between a positive cardiorespiratory outcome in the GXT and the exercise training regimen of firefighter applicants. The statistical analyses further inferred the impacts of physical characteristics on several physiological parameters. Each relationship is presented in the following paragraphs and discussed in detail with support, when applicable, from our statistical analysis.

## Physical characteristics

The height of applicants was essentially identical between groups whereas SG were on average three years younger. SG also included the youngest applicant while UG included the oldest applicant. Although the age variable was statistically different, a delta of three years between groups was not practically significant in this case as confirmed by a small effect size (Fig. 1). In addition, SG was statistically lighter than UG. When the maximal $\dot{V}O_2$ value is expressed relative to body mass, it is to be expected that applicants with lower body mass should reach higher $\dot{V}O_2$ values given the same absolute $\dot{V}O_2$ values (*Basset & Boulay, 2000*). Still, a higher body mass does not always convert into a lower $\dot{V}O_2$ value. Applicants with a large skeletal muscle mass would exhibit high maximal absolute $\dot{V}O_2$ values that would compensate for the negative body mass effect on their relative $\dot{V}O_2$ values. Inversely, applicants with a high fat mass would negatively impact their maximal relative $\dot{V}O_2$ value even though their absolute $\dot{V}O_2$ value is within normal range; a condition that may translate into an unsuccess in the GXT. The effect size computation confirmed that a difference in body mass of 7.6 kg between SG and UG had a detrimental medium effect with CI ranging from trivial up to large effect size (Fig. 1). Firefighter applicants typically

possess a greater skeletal muscle mass than the general population. This is primarily due to the resistance training they partake in to succeed in their occupational-specific physical test. In fact, the applicants need to include resistant training to challenge the tasks related to a job-specific simulation as part of the recruitment process. This anthropometric factor adds to the set of physical assets applicants should display to be successful in a firefighting career. Since skeletal muscle mass of applicants was not collected, it is therefore difficult to speculate on the body composition of heavier applicants in our sample. Nevertheless, scrutinizing data from applicants with body mass $\geq$ 100 kg, 7/10 applicants failed the GXT and they all had an absolute maximal $\dot{V}O_2$ value below the sample mean; outcomes suggesting lower muscle mass along with higher fat mass compared to applicants who were successful.

## Physiological parameters

Some may argue that successful applicants had higher maximal $\dot{V}O_2$ values because they exerted themselves more during the GXT. This was not the case with our sample because both groups reached similar maximal HR, RER, and breathing frequency, which are indicators of maximal effort. Mean maximal $\dot{V}O_2$ value for our sample was 46.5 $\pm$ 6.4 ml kg$^{-1}$ min$^{-1}$, which fits with a previous research study on firefighters that found a mean of 46.6 $\pm$ 6.0 ml kg$^{-1}$ min$^{-1}$ (*Donovan et al., 2009*). On average, the maximal $\dot{V}O_2$ value for SG was 28% higher as compared to UG. Since the cutoff to divide the two groups was based on their maximal $\dot{V}O_2$ values, maximal absolute $\dot{V}O_2$, $\dot{V}CO_2$, and $\dot{V}_E$ values were also expected to be higher in the group showing a more desirable cardiorespiratory fitness level (Table 1).

Heart rate recovery is defined as the rate at which HR decreases within the following minutes after the cessation of physical exercise and reflects the dynamic balance between parasympathetic reactivation and sympathetic withdrawal (*Qiu et al., 2017*). It is a simple assessment of the autonomic nervous system function indicating one's ability to adjust HR to match changes in exercise demands. Accordingly, HR recovery is widely used as a guide to monitor changes in physical fitness and training status. Previous research demonstrated a strong relationship between HR recovery and $\dot{V}O_{2max}$ among elite athletes (*Suzic Lazic et al., 2017*). More precisely, the authors reported that HR recovered on average 58 bpm 2-minute after the cessation of the GXT. In our analysis, mean HR were on average 42 and 37 bpm lower 2-minute after the cessation of the GXT among applicants in the SG and UG, respectively. It is, thus, not surprising that HR recovery was considered a beneficial parameter of success displaying a medium effect size with CI ranging from trivial up to large effect size (Fig. 1).

## Exercise training regime

As above-mentioned, firefighters generally possess a higher skeletal muscle mass than the general population because muscular strength is required for their job. As a result, the physical preparation for applicants should ideally include regular endurance exercise and strength exercise training. Surprisingly, endurance exercise showed a wide range of training hours among all applicants, varying from 0 to 12 h/week and, although, no endurance or

total exercise training systematically indicates a failure in the GXT, on average SG trained 75 more minutes (53%) each week in endurance exercise than UG. This is equivalent to an extra five hours of endurance exercise training each month, a behavior that should bring about a higher cardiorespiratory fitness level after a few months of additional training. Total self-reported exercise was also higher in SG as compared to UG. In the same trend, SG added on average 46 more minutes of strength exercise per week compared to UG, even though it did not translate into a statistically significant difference between groups (displaying a small effect size with half of the CI in the trivial area (Fig. 1)). Therefore, it was inferred that both groups weight trained similarly. Overall, successful applicants exercised 40% (119 min) more each week in TE. Likewise, this additional eight hours of training volume accumulates every month and should display a favorable effect over time on the cardiorespiratory fitness level of applicants as a consequence of concurrent training effect (*Dudley & Fleck, 1987*).

In addition, our results showed that applicants in the yellow zone (40–45 ml min$^{-1}$ kg$^{-1}$) were not visually and statistically different from those in the red or green zones for all parameters. However, body mass and HR recovery were different between the red and green zones. Figure 2 shows a difference in medians of 10 kg for body mass between applicants in the red zone and those in the green zone which is a substantial difference whereas HR recovery was 8 bpm higher in the green zone compared to those in the red zone. Moreover, applicants in the green zone displayed higher endurance exercise and total exercise training times as compared to those in the red zone (Fig. 3). This can be visually confirmed by the notches between the red and green zones that do not overlap, indicating a distinct difference in the medians between the two zones. The median for endurance exercise in the green zone was equivalent to 4 h/week, 95% CI [3–4.5] and the median for total exercise was 7 h/week, 95% CI [6–8]. Although it does not guarantee success, aiming to train similarly to applicants in the green zone would most likely lead to a higher probability of succeeding in the GXT than replicating the amount of exercise training performed by the group in the red zone. In fact, 82% of all applicants who trained ≥ 7 h/week in total exercise were successful and 87% of all applicants who trained ≥ 4 h/week in endurance exercise reached the standard in the GXT, reinforcing the importance of endurance-based activities in the exercise training regimen. Performing 4 h/week of endurance type of exercises (no less than 3 h/week) within a training volume of 7 h/week (no less than 6 h/week) on a regular basis is well within the capacity of many firefighter applicants and hopefully exercise training will become part of their lifestyle throughout their careers.

One of the most significant aspects of the manuscript relates to the concurrent training or cross-training effects. In fact, the chronic response brought about by 4 h of endurance exercise performed per week by the successful applicants were clearly not hindered by the 3 h of strength training. These outcomes partly concur with *Coffey et al. (2017)* who report that chronic response to concurrent training depends on individual's fitness status, the fitter the individual, the greater the disruptive impact of strength training on cardiorespiratory fitness. However, the same authors mentioned that low to moderate intensity and volume of training do bring no interference effect of strength on cardiorespiratory response. They

further suggest that the potential of resistance training to increase oxidative metabolism might contribute to the cardiorespiratory fitness of recreationally active individuals such as our firefighter applicants. Somehow, they agree with *Dudley & Fleck (1987)* and *Loy, Hoffmann & Holland (1995)* reporting that concurrent training does not affect the development of oxidative capacity. Those conclusions and our outcomes further confirm the benefit of strength training on cardiorespiratory fitness in moderately fit individuals and that the total volume of training represents a very good indicator of cardiorespiratory fitness in moderately trained individuals.

## CONCLUSIONS

Physical exercise is an important aspect of preparation for fitness testing among firefighter applicants and it usually encompasses endurance and strength type of exercise training. The main objective of the current secondary analysis was to determine if there was any difference in exercise training routine between successful and unsuccessful applicants and, if so, to provide practical guidance to future applicants preparing for their GXT. Based on a 5.6% total variability, it was determined that a good target for applicants would be to aim for a maximal $\dot{V}O_2$ value $\geq 45$ ml $kg^{-1}$ $min^{-1}$ that correspond to the green zone. While it does not guarantee success in the GXT, future firefighter applicants are encouraged to replicate the exercise training regime self-reported by successful applicants in the green zone. This suggestion is even more important for those who currently have a maximal $\dot{V}O_2$ value $\leq 40$ ml $kg^{-1}$ $min^{-1}$. Our results showed endurance exercise and total exercise differences between SG and UG but no statistical difference in strength exercise between groups. This does not mean that strength has no effect on success in the GXT, but rather that we cannot reject the null hypothesis with our sample size. Thus, future applicants are encouraged to perform strength exercise regularly to be ready physically for their occupational specific test, which requires a significant amount of muscular strength. However, if someone must choose between the two types of exercise training, our findings suggest that endurance exercise should be favored. In short, future firefighter applicants should embrace the following practical recommendations as guidance to prepare for the GXT as part of their recruitment process:

- 7 h/week of total exercise training, from which 4 h/week should be endurance-based exercise training;
- Heart rate recovery at 2-minute $\geq 43$ beat per minute; and
- Healthy body mass composition with optimal muscle mass and fat mass.

### Funding
The authors received no funding for this work.

### Competing Interests
The authors declare there are no competing interests.

## Author Contributions

- Sylvie Fortier conceived and designed the experiments, performed the experiments, analyzed the data, prepared figures and/or tables, authored or reviewed drafts of the article, and approved the final draft.
- Liam P. Kelly conceived and designed the experiments, performed the experiments, authored or reviewed drafts of the article, and approved the final draft.
- Fabien A. Basset conceived and designed the experiments, authored or reviewed drafts of the article, dr. Basset was the supervisor of Fortier and Kelly master thesis, and approved the final draft.

## Human Ethics

The following information was supplied relating to ethical approvals (i.e., approving body and any reference numbers):

Memorial University of Newfoundland granted Ethical approval to carry out the within its facilities (Human Investigation Committee Ethical Application #HIC-10,201).

## Data Availability

Data and code are available at GitLab: https://gitlab.com/sfortier/peerj/-/tree/4343c0aa2f6d38f6099c6cb74f24d732126c3041/.

## Supplemental Information

Supplemental information for this article can be found online at http://dx.doi.org/10.7717/peerj.13832#supplemental-information.

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
