# Peer review of "Practical guidance for firefighter applicants preparing for cardiorespiratory fitness testing: a secondary analysis of self-reported physical activity levels"

_PeerJ, doi:10.7717/peerj.13832_

## Round 0.1 · original submission · Major Revisions

Dear authors, your study was well designed and well written. I have received feedback from the three reviewers and based upon their comments you need to make major revisions to your manuscript. Please review the comments thoroughly and amend your manuscript accordingly. I look forward to receiving your amended manuscript (and responses to reviewers) in a timely manner.

Additionally, your supplemental data is incomplete Thanks, A/Prof Mike Climstein

·

Basic reporting

Congratulations to the authors for a well written study. The references used are relevant and appropriate and sufficient background has been provided. The article is well structured with appropriate information provided. The figures are clear and the raw data has been shared.

The introduction highlights the importance of cardiorespiratory fitness within this population and cites a relevant article of Canadian Firefighters which is where the rationale for the minimum criteria of 42.5ml/kg/min. The introduction then links to both cardiovascular issues in firefighters and also successful completion of the GXT, which appears to be a barrier test for application for entry into a career in firefighting in Canada. What could be improved in the introduction is that the GXT is one single part of the entry testing and one occupationally relevant physical attribute. The article by Gledhill and Jamnik highlights that strength is also critical in this population, with lifting, carrying and pulling objects being deemed a commonly encountered physical demand of firefighters. In addition to the GXT, applicants will have to complete additional occupational specific tests including victim drags, ladder lifts, hose drags, sled pulls and equipment carries as part of the York, Brock or CFFM FPFE assessments. The requirements of strength, power and muscular endurance should not be dismissed. While the authors do elude to the importance of muscular strength in the discussion, it could also be highlighted in the introduction, that while aerobic fitness is important, it is one of many attributes required. While I understand that the intent was to explore training history and GXT success, the fact that the GXT is one entry test of many should also be highlighted.

Table 1 needs a footnote, reminding the reader of the abbreviations.

A quick recap of the aims of the study and general findings may be appropriate a the beginning of the discussion as opposed to jumping straight into the physical characteristics.

Experimental design

The article is original research which is well suited to the journal. The research question is well defined, relevant and meaningful and the research will assist in preparation for entry into the fire service. The articles technical and ethical standard and high and the study is reproducible.

While the study mentions numerous times that it is a secondary analysis, it is not clear what it is a secondary analysis of. Likewise, I believe the GXT can be performed at numerous different locations prior to applying to the fire service. Was this analysis just the data collected at the lab of the authors? I would assume that candidates would present with their results from an external qualified organisation prior to progressing through the recruitment process. So I am also making the assumption that these 121 are the ones who were testing at York University?

The protocol for the GXT is well described and the measurement technique is provided. Do the authors have any reliability or accuracy measurements of the Oxycon pro system?

Validity of the findings

This is an under researched area, and data is needed for both applicants to the fire service and the fire service itself. Thank you to the authors for providing their data set. The addendum 2 is comprehensive and the steps taken to omit self-reported data which does not appear feasible is logical. In addition, differentiating between activity and exercise is appropriate. The statistical method is well described and appropriate. Conclusions are well stated and agree with previously published research within the fire service. Having an adjustment for daily variability was also a good idea.


Should someone wish to use the provided dataset, which I would encourage in this population given the scarce amount of research and difficulty with access at times, I would advise the authors to provide some further clarity. Although it appears that the nine observations were considered as outliers using the z scores, age, body mass and absV02 were retained. There are nine entries listed as 0 in the data set for EE, SE and TE. It would be helpful to differentiate between those which were subjectively reported as a 0 current exercise, vs those who were recoded as a non-entry/outlier. For example, while ID 51 with a relative V02 of 30.7ml/kg/min may report 0 total exercise which would match their aerobic fitness level, I would assume ID 120 with an relative V02 of 54.5ml/kg/min would be one of the ones who erroneously entered data. Perhaps a zero score for those who reported no exercise and a blank for those who were considered outliers would help this.

The blank scores in VC02, VE, BF, RER for ID113 and ID68 are eluded to in the addendum however was there a reason for these values being missing? Was it equipment failure?

Additional comments

Thank you for the opportunity to review this paper. Congratulations to the authors on a well conducted and written study. This is an area in need of more research to provide valuable information to our first responders.

Reviewer 2 ·

Basic reporting

.

Experimental design

.

Validity of the findings

.

Additional comments

The study and paper are fine and the paper is publishable as it is. However, somewhere between 20% to as much as 40% of fitness may be genetically determined, independent of physical activity and exercise training, which should me mentioned and factored in.

·

Basic reporting

This manuscript is well written and easy to read. The main suggested improvements are in the referencing used in the discussion. As written, there are only 3 references and I do not believe this is currently sufficient. Therefore, it does not demonstrate how the study fits into the broader knowledge of the field. I have attached additional comments to assist the author in improving the manuscript

Experimental design

The study is well designed and executed. My main concern is on the justification of standard setting of 42.5ml.kg.min. This justification needs to be clearly articulated in the manuscript. It is also necessary for the authors to justify or explain why an unweighted running test is currently used as the fire fighter fitness standard. See attached document for further comments.

Validity of the findings

As mentioned, the benefit to the literature is ambiguous. This can be improved with further discussion. The conclusion is clear and relates to the results however I think the exercise recommendations are better suited in the discussion.

Additional comments

This is a novel study and will help many firefighter applicants. Please see the attached document for specific comments,

---

## Round 0.2 · accepted · Accept

Dr Basset and colleagues, thank you for addressing all of the Reviewers' comments and concerns. I am pleased to inform you that I am therefore pleased to recommend your amended manuscript for publication in PeerJ. Thank you for supporting PeerJ and we look forward to your future submissions. A/Prof Mike Climstein